

# Data-based investigation of the effects of canopy structure and shadows on chlorophyll fluorescence in a deciduous oak forest

Hamadou Balde[1,2,3,4], Gabriel Hmimina[1], Yves Goulas[1], Gwendal Latouche[2], Abderrahmane Ounis[1], Kamel Soudani[2]

[1]Laboratoire de Météorologie Dynamique, Sorbonne Université, IPSL, CNRS/L'École polytechnique, 91128, Palaiseau Cedex, France

[2]Ecologie Systématique et Evolution, Université Paris-Saclay, CNRS, AgroParisTech, 91190, Gif-sur-Yvette, France

[3]Centre national d'études spatiales (CNES), 18 av. Edouard Belin, 31400 Toulouse

[4]ACRI-ST, 260 Route du Pin Montard, BP 234, 06904 Sophia-Antipolis, France

*Correspondence to*: Hamadou Balde (hamadou.balde@lmd.ispl.fr)

**Abstract:** Data from satellite, aircraft, drone, and ground-based measurements have already shown that canopy scale sun-induced chlorophyll fluorescence (SIF) is tightly related to photosynthesis, which is linked to vegetation carbon assimilation. However, our ability to effectively use those findings are hindered by confounding factors,

including canopy structure, fluctuations in solar radiation and in sun-canopy-geometry that highly affect the SIF signal. Thus, disentangling these factors has become paramount in order to use SIF for monitoring vegetation functioning at canopy scale and beyond. Active chlorophyll fluorescence measurements ($F_{yieldLIF}$), which directly measures the apparent fluorescence yield, have been widely used to detect physiological variation of the vegetation at leaf scale. Recently, the measurement of $F_{yieldLIF}$ has become feasible at the canopy scale, opening up new

opportunities to decouple structural, biophysical, and physiological components of SIF at the canopy scale. In this study, based on top-of-canopy measurements above a mature deciduous forest, reflectance (R), SIF, SIF normalized by incoming photosynthetically active radiation ($SIF_y$), $F_{yieldLIF}$, and the ratio between $SIF_y$ and $F_{yieldLIF}$ (named $\Phi_k$) were used to investigate the effects of canopy structure and shadows on the diurnal and seasonal dynamics SIF. Further, random forest (RF) models were also used to not only predict $F_{yieldLIF}$ and $\Phi_k$, but also

provide an interpretation framework by considering additional variables, including the R in the blue, red, green, red-edge, and near-infrared bands, SIF, $SIF_y$, and sun zenith (SZA) and azimuth (SAA) angles. Results revealed that the SIF signal is highly affected by the canopy structure and sun-canopy geometry effects compared to $F_{yieldLIF}$. This was evidenced by the weak correlations obtained between $SIF_y$ and $F_{yieldLIF}$ at the diurnal timescale. Furthermore, the daily mean $\overline{SIF_y}$ captured the seasonal dynamics of daily mean $\overline{F_{yieldLIF}}$ and explained 58% of

its variability. The findings also revealed that reflectance in the near-infrared (R-NIR) and the $NIR_v$ (the product of NIR by the normalized difference vegetation index) are good proxies of $\Phi_k$ at the diurnal timescale, while their correlations with $\Phi_k$ decrease at the seasonal timescale. With $F_{yieldLIF}$ and $\Phi_k$ as outputs and the abovementioned variables as predictors, this study also showed that the RF models can explain between 86% and 90% of $F_{yieldLIF}$, and 60% and 70% of $\Phi_k$ variations under clear sky conditions. In addition, the predictor importance estimates for

$F_{yieldLIF}$ RF models revealed that R at 410, 665, 740, and 830 nm, SIF, $SIF_y$, SZA, and SAA emerged as the most useful and influential factors for predicting $F_{yieldLIF}$, while R at 410, 665, 705, and 740 nm, SZA, and SAA are crucial for predicting $\Phi_k$. This study highlighted the complexity of interpreting diurnal and seasonal dynamics of SIF in forest canopies. These dynamics are highly dependent on the complex interactions between the structure of



the canopy, the vegetation biochemical properties, the illumination angles (SZA and SAA) and the light conditions

(ratio of diffuse to direct solar radiation). However, such measurements are necessary to better separate the variability in SIF attributable to radiation and measurement conditions from the subtler variability attributable to plant physiological processes.

## 1. Introduction

Spatial and temporal information on vegetation status are crucial to gain a better understanding of vegetation

functioning and productivity. Remotely sensed data mostly from satellite and airborne platforms have provided such information for decades now (Ustin and Middleton, 2021). However, most of the remote sensing methods used for detecting and monitoring the dynamics of vegetation properties were exclusively based on vegetation greenness derived from optical vegetation indices (VIs), such as the normalized difference vegetation index (NDVI), and more recently the near-infrared reflectance of vegetation index ($NIR_v$), which have been broadly and

successfully used to estimate some biophysical and biochemical attributes, including leaf area index (LAI), fraction of absorbed photosynthetically active radiation (fAPAR), and leaf chlorophyll content (Campbell et al., 2019; Zeng et al., 2022b).

Sun-induced chlorophyll fluorescence (SIF) is a direct indicator of the vegetation photosynthetic activity that responds to abiotic stresses, such as heatwaves and droughts, earlier than VIs (Frankenberg et al., 2011; Guanter

et al., 2014; Rascher et al., 2015; Jonard et al., 2020). Further, SIF is not directly impacted by soil background as green vegetation is the only source of chlorophyll fluorescence in the red and far-red. The potential carried by SIF is currently used for estimating and monitoring terrestrial gross primary productivity (GPP) across different vegetation types, including, crops, deciduous forests, evergreen forests, tropical forests, wetlands, etc. (Li and Xiao, 2022; Verma et al., 2017; Wood et al., 2017; Balde et al., 2023), for assessing vegetation structural changes,

and estimating crop productivity (He et al., 2020; Liu et al., 2022).

However, because of the coarse spatial scale of the satellite products used in these above mentioned studies, the results are inconclusive and it is still questioned whether SIF can provide reliable estimates of GPP at different spatial scales and temporal resolutions across different vegetation types, and more particularly under various abiotic stress conditions (Paul-Limoges et al., 2018; Yazbeck et al., 2021; Lin et al., 2022; Balde et al., 2023; Sun

et al., 2023b). Further, satellite SIF signals are also subject to the effects of the interactions between the roughness of upper canopy layers (tree forms, gaps), and the solar zenith (SZA) and azimuth (SAA) angles. These interactions modulate the spatial and temporal distributions of sunlit and shaded leaves, the light distribution within the canopy and the main physiological processes, such as photosynthesis, evapotranspiration, and stomatal conductance (Gao et al., 2022; Morozumi et al., 2023).

The recent increased availability of diurnal and seasonal time series of SIF data from airborne, drone, and ground-based measurements was crucial for gaining a better understanding of what drives SIF at various spatial and temporal scales and across biomes (Damm et al., 2015; Rascher et al., 2015; Yang et al., 2017; Goulas et al., 2017; Wang et al., 2021; Zhang et al., 2021; Wang et al., 2022; Xu et al., 2021; De Cannière et al., 2022). However, interpretation of locally measured SIF data should be cautiously carried out. In fact, rapid variations in fluorescence

may be due to local effects linked to the conditions of illumination and to the light absorption by the canopy. These effects may lead to significant variations in SIF without substantial variations in photosynthesis of the entire




canopy. Therefore, distinguishing the effects of endogenous factors related to canopy structure from the effects of photosynthesis changes on SIF signal is warranted.

At the top-of-canopy, the radiative transfer of SIF can be resumed within Eq. (1):

$$SIF = PAR \times fAPAR \times \Phi_F \times f_{esc} \qquad (1)$$

where $PAR$ is the incoming photosynthetically active radiation (400-700 nm), which is the first driver of canopy SIF signal (Miao et al., 2020). $fAPAR$ is the fraction of absorbed $PAR$ by the vegetation, and $f_{esc}$ is the fraction of all chlorophyll fluorescence photons emitted from all leaves and escaped from the canopy, also known as fluorescence escape probability fraction, which is dependent on the biophysical and biochemical properties of the canopy and on the sun and view geometry. $\Phi_F$ is the chlorophyll fluorescence quantum yield (the ratio of the total

amount of photons emitted to the total amount of photons absorbed by the chlorophyll pigments) and hence it is a direct indicator of the photosynthetic efficiency. From Eq. (1), it is explicit that in order to interpret top-of-canopy $SIF$ and use it as a proxy of $\Phi_F$ and photosynthesis, it is necessary to understand and disentangle $\Phi_F$ from the $SIF$ canopy structure dependent variations (due to the spatiotemporal effect's variations in sunlit and shaded leaves and to the light distribution and attenuation within the canopy) that are contained in $fAPAR$ and $f_{esc}$.

Disentangling the photosynthetically dependent variations from the canopy dependent ones in SIF signal is critical to use SIF as a proxy of vegetation response to changing environmental conditions and to abiotic stresses at large scales. It is especially needed for the upcoming Fluorescence Explorer (FLEX) satellite mission that aims at providing measurements of SIF at its full spectral emission (670-780 nm) and with unprecedented spatial resolution

(300 m) and repeated global coverage (Drusch et al., 2017). Therefore, the top-of-canopy SIF measured together with GPP at the carbon flux sites can play a substantial role for calibrating and validating FLEX products and airborne campaigns measurements.

Recent studies have developed novel approaches based on theoretical insights to correct SIF signal for multiple scattering and reabsorption effects (Zeng et al., 2019; Yang and van der Tol, 2018; Yang et al., 2020) by

determining the $f_{esc}$ and allowing the downscaling of SIF emission from canopy to fluorescence emission yield (Lu et al., 2020). This assumes that the entire canopy acts like a big leaf, with unique absorption, fluorescence, and photosynthetic properties. In this situation, $f_{esc}$ is the ratio of top-of-canopy SIF to SIF total and it is closely related to canopy structural variations, including LAI, leaf angle distribution, reabsorption, and sun-canopy geometry, and varies across time and space (Zeng et al., 2019). Recently, $f_{esc}$ has been estimated using $NIR_v$ or the fluorescence

correction vegetation index (FCVI). The former considers soil background effects and is the product of NDVI and the reflectance in the near-infrared (NIR) (Badgley et al., 2017) and it has successfully been used to assess photosynthesis productivity (Mengistu et al., 2021). The latter, FCVI, is framed as the difference between the NIR and the broadband visible reflectance (400-700 nm), considering that the reflectance is measured in the same direction as the SIF observations (Yang et al., 2020). Both approaches have shortcomings, as they cannot be

universally applied, because some steps in the estimation of $f_{esc}$ using $NIR_v$ are inconsistent with the radiative transfer theory (Yang et al., 2020) and their effectiveness might be greatly compromised for SIF at the red band where the scattering is much weaker than in the near-infrared. The use of FCVI is also limited as it is not suitable in sparse vegetation canopies and its computation requires hyperspectral data in the visible spectral range.

If one would like to disentangle the radiation and vegetation structure dependent SIF variations from the

physiological information in the SIF signal, determining $\Phi_F$ is required. $\Phi_F$ can be defined at the leaf scale, or even at lower scales (chloroplasts) where the absorbed light energy is dissipated following three pathways:



photosynthesis, fluorescence, and heat dissipation. Estimating leaf-scale $\Phi_F$ from canopy SIF measurements is an ongoing work that is under exploration. In addition, the computation of total absorbed photosynthetically active radiation (APAR) requires measurements of the incident, transmitted, and reflected PAR, which cannot be

measured at satellite and airborne platforms, and are not always available for all ground sites even those belonging to major carbon flux observation networks, such as the Integrated Carbon Observation System (ICOS). This is the reason why for decades the apparent $\Phi_F$ was estimated by normalizing the top-of-canopy SIF signal converted in quanta energy by the incident PAR (Daumard et al., 2012; Goulas et al., 2017). Recently, two promising approaches have been used by Zeng et al. (2022a) and Loayza et al. (2023) to estimate $\Phi_F$. To determine $\Phi_F$ over

cropped fields, including, rapeseed, barley, corn, wheat, and sugar beet, Zeng et al. (2022a) normalized canopy SIF by the near-infrared radiance of vegetation index (rNIR$_v$, the product of NDVI and the reflected vegetation radiance in the near-infrared), while Loayza et al. (2023) used the integrated vegetation reflected radiance between 500 and 700 nm on potato crop. These approaches have advantages because the effects of canopy structure and sun-canopy geometry on $\Phi_F$ estimates may be fully cancelled out, the PAR is not needed as an input, and their

applicability at the satellite scale is highly feasible. However, how much these methods are reliable and effective on estimating $\Phi_F$ under varying environmental conditions and across diverse spatiotemporal scales and vegetation types is not well explored yet.

Luckily, chlorophyll fluorescence can be measured using active methods that allow direct evaluation of the physiological status of the vegetation at the leaf and canopy scales (Porcar-Castell et al., 2014; Moya et al., 2019;

Loayza et al., 2023). In active techniques, a modulated source of light is used to excite the chlorophyll that fluoresces in the spectral range between 650 and 800 nm. For instance, the pulse amplitude-modulated techniques, which use a measuring pulsed light and an actinic continuous light, has been widely used at the leaf scale to provide direct chlorophyll fluorescence yield measurements, allowing the evaluation of photosynthesis and vegetation responses to abiotic factors for decades (Baker, 2008; Magney et al., 2017). But, its applicability at canopy and

ecosystem scales were hindered by the techniques limitations (Ounis et al., 2001). In the last decades, this gap was filled based on the use of either lasers (or laser diodes), or light emitting diodes (LED) providing short pulses of light (microsecond to even picosecond), together with a synchronized detection to measure chlorophyll fluorescence under daylight conditions at the canopy scale with in-situ or airborne remote sensing instruments (Moya et al., 2019; Ounis et al., 2016; Loayza et al., 2023). Therefore, the fluorescence efficiency can be directly

observed at the canopy and landscape scales, which is useful to gain a better understanding of terrestrial vegetation functioning. Indeed, LED induced chlorophyll fluorescence ($F_{yieldLIF}$) is less affected by the temporal and spatial (horizontal and vertical) distribution of sunlit and shaded leaves on the upper surface and within the canopy compared to SIF, but it may be highly sensitive to environmental conditions, such as heavy wind speeds (Lopez Gonzalez, 2015).

In forest stands, such as temperate deciduous forests, when the vegetation green-up and senescence phases are excluded, LAI is merely constant. However, the spatial dynamics in LAI may be large from one plot to another. Thus, the canopy structural effect correction on SIF signal is all the more crucial from a spatial view point. Further, SIF signal is subject to diurnal variations due to the complex interactions between lighting conditions (diffuse/total radiation, solar and viewing angles) and canopy structure (Aasen et al., 2019; Xu et al., 2021). Therefore, correcting

SIF from these effects, which are very local, is warranted for (i) interpreting and upscaling SIF signal spatially and temporally across diverse vegetation types, (ii) disentangling the physiological response from variations due to





exogenous effects on SIF, (iii) assessing how SIF responds to extreme environmental conditions (heatwaves, drought, etc.), and ultimately (iv) gaining a better understanding of the relationships between SIF and GPP. Nevertheless, to the best of our knowledge, an attempt to use active fluorescence measurements at the canopy scale

to correct SIF from canopy structure, incident sunlight, and sun-canopy geometry effects has not been addressed yet.

The main objective of this work is to use active chlorophyll fluorescence ($F_{yieldLIF}$) as a reference measurement and to compare it to SIF yield ($SIF_y = SIF/PAR$) in order to analyse and correct the effects of canopy structure and sun-canopy geometry on top-of-canopy SIF at diurnal and seasonal timescales in a temperate deciduous forest,

considering diverse environmental conditions. More specifically, this study intended to (i) evaluate the relationship between $F_{yieldLIF}$ and $SIF_y$ and evidence the effects of canopy structure and sun-canopy geometry on top-of-canopy SIF through their influence over this relationship; (ii) examine these effects with the aim of developing a correction method based on reflectance measurements and lightning conditions (solar angles, ratio of diffuse to total radiation, etc.).

**2.    Materials and Methods**

**2.1.    Study site description**

This study was conducted at the Fontainebleau-Barbeau forest site (FR-Fon), which is an Eddy Covariance (EC) flux observation site belonging to the ICOS network (Delpierre et al., 2016). The site is located 53 km southeast of Paris, France. It is occupied by a temperate deciduous broad-leaf forest type. The dominant forest overstory

consisted of mature sessile oak trees (*Quercus petraea (Matt.) Liebl*), accounting for 79% of the basal area (Maysonnave et al., 2022), with an understory of hornbeam (*Carpinus betulus L.*) (for more details: http://www.barbeau.universite-paris-saclay.fr/). The stand height is around 25 m. The soil is an endostagnic luvisol, covered by an oligo-mull humus (Maysonnave et al., 2022). The climate is temperate and characterized by an annual average rainfall of approximately 680 mm and an average air temperature of approximately 11°C

(Soudani et al., 2014). The LAI is approximately 5.8 $m^2.m^{-2}$ using the litter collection method over the 2012-2018 period (Soudani et al., 2021). At the Fontainebleau-Barbeau site, carbon and water fluxes have been continuously monitored at 35 m height using the EC method. The main micrometeorological variables, including incident and reflected radiations, are measured at high frequency (1 min), while vapor pressure deficit, precipitation, air and soil temperature, water table depth, soil moisture, and wind speed are either recorded or estimated at a half-hourly

timescale.

**2.2.    Sun-induced and light-emitting diode induced chlorophyll fluorescence, and reflectance measurements of the canopy**

**2.2.1.    Sun-induced chlorophyll fluorescence measurement system**

In the framework of the ECOFLUO project, a passive in-situ spectral measurement automated instrument (named

SIF3) was developed based on a collaboration between the "Laboratoire de Météorologie Dynamique (LMD), École Polytechnique, France et Laboratoire Écologie, Systématique et Évolution (ESE), Université Paris-Saclay, France". SIF3 acquires continuous measurements of incident and reflected radiation above the canopy. It was installed at the top of the 35 m height tower of Fontainebleau-Barbeau site above the canopy in July 2021



(Supplementary materials Figure S1). To avoid artificial shading of the measured area, SIF3 was set to the southern

part of the tower.

The SIF3 measurement system includes a control computer (LattePanda V1, LattePanda Shanghai, China and two Arduino microcontrollers), two spectrometers with coolers, shutter controllers, a general cooler with temperature controller inside the box, two optical fibers, a reference panel, a servo motor, a PAR sensor, a GPS, temperature and relative humidity sensors, and a camera. The two spectrometers are a high-resolution spectrometer (ASEQ

instruments, Vancouver, Canada, HR1-T model with thermoelectric cooling) and a broad band spectrometer (ASEQ, LR1-T model with thermoelectric cooling). The high-resolution spectrometer (HR1-T) has a spectral range between 650 and 800 nm, a high spectral resolution with full width at half maximum (FWHM) of approximately 0.3 nm. The HR1-T was used to determine sun-induced chlorophyll fluorescence. The broad band spectrometer (LR1-T) has a spectral range between 400 and 1000 nm and a FWHM of approximately 1.5 nm. It

was used to measure canopy reflectance and the optical vegetation indices (VIs).

In order to reduce the noise and dark current, both spectrometers were contained within a dry and thermoregulated box system that maintained the temperature at $19 \pm 0.61$ °C. SIF3 performs sequential vegetation reflected radiance measurements and irradiance measurements on a polytetrafluoroethylene (PTFE) reference panel (PMR10P1, Thorlabs, Maisons-Laffitte, France). The sequential measurements of SIF3 were: first to measure the reference

PTFE with LR1-T and HR1-T spectrometers, and second to measure vegetation reflected radiance with both spectrometers. Within one measurement of the target canopy or the reference, each spectrometer performed the following steps: (i) optimizing the integration time (IT) for measurement, (ii) measurement, and (iii) measurement of the dark current. The date and time at the start and end of each measurement were recorded. Two 15 m long optical fibers (FT1000EMT and FT1000UMT, Thorlabs, Maison-Laffitte, France, for HR1-T and LR1-T

spectrometers, respectively) with a 1000 µm core diameter and a numerical aperture of 0.39 NA were used to measure the irradiance of the reference and the radiance of the canopy, at the nadir position. The field-of-view (FOV) of each measuring channel is adjusted to 25° with the use of a Gershun tube to ensure a flatter spatial response and covered approximately 6 m² of the canopy area. Long-pass optical filters (5CGA-550, cut-off wavelength 550 nm and 5CGA-375, cut-off wavelength 375 nm, Newport, Irvine, CA, USA, for the HR1-T and

LR1-T channels, respectively) were placed in front of each tube to avoid second order detection and to protect fiber ends. The dark current measurements were subtracted from the reference and canopy measurements before SIF retrieval. The IT of each spectra was automatically optimized to achieve values that are as high as possible, but unsaturated to improve as much as possible the signal-to-noise ratio (SNR). Note that SIF3 integrates a quantum sensor to measure the PAR at high frequency and a camera that allows taking RGB images of the canopy

in the FOV. Before the installation of SIF3 in the field, we performed lens alignment, radiometric and spectral calibrations of the instrument using a calibrated light source (4P-GPS-060-SF and EHLS-100-075R, Labsphere, North Sutton, NH, USA).

### 2.2.2.   Light-emitting diode induced chlorophyll fluorescence measurement system

$F_{yieldLIF}$ measurements were acquired with an active fluorometer instrument, named LIF, developed in the LMD

laboratory, which was installed at the top of the 35 m height tower next to SIF3 above the canopy. This instrument is very similar to the one described by Moya et al. (2019). It uses a blue LED array (ENFIS Ltd, Swansea, UK; peak wavelength 455 nm, FWHM 21 nm, radiant power 6 W) as an excitation source. To separate the chlorophyll



fluorescence emission induced by the LED from that induced by daylight and from the reflected sunlight in the filter bandwidth, the LED light is pulsed at a variable frequency with a pulse duration of about 5 µs. Note that the instrument uses a bimodal excitation conditioned by the PAR: for PAR < 90 µmol m$^{-2}$ s$^{-1}$ (night time), the frequency is set at 30 Hz, while it is set at 200 Hz (daytime) for PAR > 100 µmol m$^{-2}$ s$^{-1}$. This bimodal excitation scheme helps to avoid variable fluorescence induction during night and to increase SNR during daytime. The instrument optical head consisted of two main parts: (i) the source module that includes the blue LED array, its electronic driver, a heat dissipation module and a Fresnel lens (diameter 180 mm) to collimate the excitation light, and (ii) a detection module that includes a second Fresnel lens of the same diameter, a set of optical filters, a large area PIN photodiode (10x10 mm$^2$, S3590, Hamamatsu Photonic, Japan) and a laboratory designed amplifier that selects the LED induced fluorescence signal (F$_{yieldLIF}$) from the reflected sunlight in the same wavelengths band (LNIR). This amplifier uses a sample and hold circuit (AD585, Analog Devices, Wilmington, MA, USA) to deliver the peak value of the fluorescence signal to the digital analog (AD) conversion card (USB 6212, NI, Austin, Texas, USA) and a lowpass electronic filter to deliver LNIR to the same card. The set of optical filters includes a highpass interferential filter with a cut-off wavelength at 400 nm to reject UV light, a second highpass interferential filter with a cut-off wavelength at 650 nm to reject the excitation light and a 3 mm thick RG9 filter (Schott, Germany) to select the far-red fluorescence emission over 725 nm. The FOV can be controlled thanks to an onboard camera (RLC-520A, Reolink, Hong-Kong). We selected a top of the canopy area in the FOV of the SIF instrument, resulting in a 9 m measuring distance with a viewing zenith angle of 30°. However, as the FOV of the instrument is about 100 mrad, the measured area was about 0.4 m$^2$, which is much smaller than the FOV of SIF3 (approximately 6 m$^2$). Power supplies as well as synchronisation and acquisition electronics are enclosed in a separate box, connected to the optical head by a 5 meters long cable. F$_{yieldLIF}$ and LNIR are stored on disk with an acquisition and control program written in LabVIEW (NI, Austin, Texas, USA) that runs on a LattePanda V1 microcomputer. Other variables such as PAR and LED, photodiode and box temperatures are also continuously monitored.

### 2.3. Canopy sun-induced chlorophyll fluorescence retrieval

As spectral measurements are recorded in digital counts, they were converted into radiometric units before SIF retrieval. SIF was retrieved at the far-red oxygen observation band (O$_2$-A) from the HR1-T canopy reflectance measurements. Data quality control is performed prior to SIF retrieval following the protocol proposed by Cogliati et al. (2015) to put aside abnormal data caused by abrupt changes in incident radiation. SIF retrieval was performed using the classical three-band Fraunhofer line discrimination (3FLD) method at O$_2$-A band (Meroni et al., 2009; Daumard et al., 2012).

The 3FLD approach is rooted in the FLD principle, which requires measurements in two channels, one inside and one outside a Fraunhofer or absorption line (Meroni et al., 2009). The FLD hypothesis is based on the consistency of reflectance and SIF at both bands. However, studies have found evidence that the two variables are not constant (Meroni et al. 2009). The 3FLD method rather assumes that reflectance and SIF vary linearly around the absorption band considered, which solves the limitation given by the FLD hypothesis, and uses three spectral bands per absorption line to retrieve SIF (Zhang et al., 2021). The 3FLD SIF retrieval at 760 nm (O$_2$-A band) can be derived as follows:

$$SIF_{760} = \frac{(E_l \times w_l + E_r \times w_r) \times L_{in} - (L_l \times w_l + L_r \times E_r) \times E_{in}}{(E_l \times w_l + E_r \times w_r) - E_{in}} \qquad (2)$$



$$w_l = \frac{\lambda_r - \lambda_{in}}{\lambda_r - \lambda_l}, \quad w_r = \frac{\lambda_{in} - \lambda_l}{\lambda_r - \lambda_l}$$

where $L$ is the upwelling radiance. $E$ is the downwelling irradiance measured on the reference panel. Indices '$r$', '$l$' and '$in$' represent the reference bands at the left, right, and inside the absorption band, respectively. $w_l$ and $w_r$ denote the weighting factors depending on the wavelength $\lambda$ on the left, inside, and on the right of the absorption band. Within this study, the left, inside and right bands were set at 757.86, 760.51, and 770.46 nm, respectively.

### 2.4. Theoretical derivations of $\Phi_k$, vegetation indices, and SIF yield

$NIR_v$ has been used to isolate vegetation signal properties from soil background and to correct canopy-scale far-red SIF for scattering effects (Badgley et al., 2017). $NIR_v$ can be computed according to (Badgley et al., 2017) and (Zeng et al., 2019) using the following Eq.:

$$NDVI = \frac{R_{[780-800]} - R_{[670-680]}}{R_{[780-800]} + R_{[670-680]}} \tag{3}$$

$$NIR_v = R - NIR_{850} \times NDVI \tag{4}$$

where $R$ represents the spectral reflectance and the index number denotes the wavelength range or wavelength at which the reflectance was measured. In Eq. (4), $NIR_v$ is largely dependent on the LAI, the leaf angle distribution, and the geometry of the sun-canopy, as well as on the influence of fluctuations in incident radiation at the diurnal and seasonal timescales.

$F_{yieldLIF}$ is an active measurement and is not directly dependent on the ambient light conditions. Thus, it is not impacted by ambient radiation changes, because the measured LED induced chlorophyll fluorescence is directly and only emitted by the leaves targeted by the LED. Variations in $F_{yieldLIF}$ are then presumably only induced by changes in the photosynthetic pigment concentrations, in the leaf area inside the FOV, and in the vegetation functioning that modulates the chlorophyll fluorescence quantum yield. As no significant phenological changes occurred during the study period, we assumed that the $F_{yieldLIF}$ is free from the vegetation structure and sun-canopy geometry effects and can be used as a reference measurement in this respect. The blue LED light can be considered as constant and therefore, from Eq. (1) we can assume that $\Phi_F$ is equal to $F_{yieldLIF}$ and then Eq. (1) becomes:

$$SIF = PAR \times fAPAR \times F_{yieldLIF} \times f_{esc} \tag{5}$$

$$\frac{SIF}{(PAR \times F_{yieldLIF})} = fAPAR \times f_{esc} \tag{6}$$

From Eq. (6), we defined $\Phi_k$ as following:

$$\Phi_k = \frac{SIF}{(PAR \times F_{yieldLIF})} = \frac{SIF_y}{F_{yieldLIF}} \tag{7}$$

Note that this is a simplification of the complex relation that does exist between $SIF_y$ and $F_{yieldLIF}$, as SIF yield and $F_{yieldLIF}$ respond differently to canopy structure effects. At the diurnal timescale, $\Phi_k$ is subjective to variations in leaf angle distribution, incident sunlight or atmosphere conditions (clear or cloudy sky conditions), and to the effects of sun-canopy geometry (including SZA and SAA).

In remote sensing, the total amount of light absorbed by the canopy cannot be directly measured. This quantity is highly dependent on the solar angle and canopy structure (distribution of light and shaded areas at the top and inside the canopy). Hence, by normalizing the canopy emitted SIF by the incident PAR, it is possible, as a first approximation and empirically, to partially disentangle the SIF signal from its dependence to incident radiation and thus to detect some changes in the vegetation properties or the plant physiological responses to abiotic factors.



Therefore, the $SIF_y$ was calculated using the PAR measured at the top of the EC tower site. Note that the SIF fluxes were converted into quanta units following (Daumard et al., 2012) prior to $SIF_y$ calculation.

$$SIF_y = SIF/PAR \tag{8}$$

## 2.5. Data analysis

In this study, we used data measurements from June to August 2022. As radiation-limited photosynthesis is expected in early morning and late afternoon, due to lower incoming irradiance, only the data recorded between 9:00 am and 16:00 pm (UTC) were considered in this study. The negative SIF values, the $SIF_y$ values higher than

315 mean $\pm$ 3 standard deviation, and the PAR data less than 200 µmol m$^{-2}$ s$^{-1}$ were excluded in the analysis. First, we applied a linear model to analyse at the daily and seasonal timescales the strength of the relationships: i) between $SIF_y$ and $F_{yieldLIF}$, and ii) between $NIR_v$ and $\Phi_k$. Note that daily means of $SIF_y$, $F_{yieldLIF}$, $NIR_v$, and $\Phi_k$ are hereafter noted $\overline{SIF_y}$ and $\overline{F_{yieldLIF}}$, $\overline{NIR_v}$ and $\overline{\Phi_k}$. The coefficient of determination ($R^2$) and the p-value are used to assess the strength of the correlations. These relations are examined at instantaneous (seconds to minutes) and daily

(averaged data from 9 am to 16 pm) timescales. Second, we used random forest (RF) models as a tool to understand $F_{yieldLIF}$ and $\Phi_k$ dynamics by comparing their predictions based on a combination of remote sensing metrics. We chose RF models because they are non-parametric models and are well adapted for predicting nonlinear and multi-parameters relationships in complex situations and foremostly highly interpretable by using metrics, such as the importance of predictor variables and partial dependence (Breiman, 2001). Several types of RF models were

designed for estimating $F_{yieldLIF}$ and $\Phi_k$. The expression of each model and its purpose are given in Table 1. We used the clear sky condition (the fraction of diffuse PAR over total PAR < 0.3) data to train the models. It is worth noting that for $F_{yieldLIF}$ predictions using all data (clear sky and cloudy conditions) or clear sky condition data alone yielded the same results (data not shown), while for $\Phi_k$, which was defined to represent the effects of canopy structure and sun-canopy geometry, only clear sky conditions were used with respect to satellite conditions of data

acquisition. To avoid the impact of correlations of predictors on the RF models' predictions, the correlations matrix between predictors was computed (Supplementary Materials Figure S2 and S3) and then the least correlated predictors were selected to train our models. All RF models were established using 200 trees and sampled with replacement based on bag fraction of 80% (80% of the data for training and 20% for testing). The out-of-bag (OOB) predictor importance estimates were determined to evaluate the contribution of each predictor. Model

performance was evaluated using the OOB coefficient of determination (OOB $R^2$) score and the adjusted coefficient of determination (adj. $R^2$) of the correlations between the test dataset and the predictions, as well as the root mean squared error (RMSE). The closest the OOB $R^2$ and adj. $R^2$ are, the better the model is able to be generalized. All RF models were run using instantaneous measurements. For SIF and reflectance data extraction, MATLAB R2021a (MathWorks, Inc., USA) was used, and Python version 3.9.1 was used for data analysis and

visualization (Sklearn, Scipy, Seaborn, Matplotlib, Pandas, and Numpy libraries).

**Table 1**. Random forest models for $F_{yieldLIF}$ and $\Phi_k$ predictions. R denotes spectral reflectance in blue (410 nm), red (530 nm and 560 nm), green (665 nm), red-edge (705 and 740 nm), and near-infrared (830 nm). SIF is the far-red sun-induced chlorophyll fluorescence at 760 nm, $SIF_y$ is the ratio of SIF over PAR, SA stands for solar angles, including solar zenith (SZA) and azimuth (SAA) angles. $F_{yieldLIF}$ is the LED induced chlorophyll fluorescence, and $\Phi_k$ is the ratio between $SIF_y$ and $F_{yieldLIF}$.

| Model name | Inputs | Outputs | Purpose |
|---|---|---|---|
|  |  |  |  |



| | | | |
|---|---|---|---|
| FY-R | $R_{410}$, $R_{530}$, $R_{560}$, $R_{665}$, $R_{705}$, $R_{740}$, $R_{830}$ | $F_{yieldLIF}$ | To test the ability of reflectances to predict $F_{yieldLIF}$. |
| FY-R-SIF | $R_{410}$, $R_{530}$, $R_{560}$, $R_{665}$, $R_{705}$, $R_{740}$, $R_{830}$, SIF | $F_{yieldLIF}$ | To test the ability of reflectances and SIF to predict $F_{yieldLIF}$. |
| FY-R-SIFy | $R_{410}$, $R_{530}$, $R_{560}$, $R_{665}$, $R_{705}$, $R_{740}$, $R_{830}$, $SIF_y$ | $F_{yieldLIF}$ | To test the effect of apparent SIF yield (SIF normalized by PAR) to predict $F_{yieldLIF}$. |
| FY-R-SA | $R_{410}$, $R_{530}$, $R_{560}$, $R_{665}$, $R_{705}$, $R_{740}$, $R_{830}$, SZA, SAA | $F_{yieldLIF}$ | To test the ability of reflectances and solar angles to predict $F_{yieldLIF}$. |
| FY-R-SIF$_y$-SA | $R_{410}$, $R_{530}$, $R_{560}$, $R_{665}$, $R_{705}$, $R_{740}$, $R_{830}$, $SIF_y$, SZA, SAA | $F_{yieldLIF}$ | To test the ability of reflectances, SIF yield, and solar angles to predict $F_{yieldLIF}$. |
| $\Phi_k$-R | $R_{410}$, $R_{530}$, $R_{560}$, $R_{665}$, $R_{705}$, $R_{740}$, $R_{830}$. | $\Phi_k$ | To test the ability of reflectances to predict $\Phi_k$. |
| $\Phi_k$-R-SA | $R_{410}$, $R_{530}$, $R_{560}$, $R_{665}$, $R_{705}$, $R_{740}$, $R_{830}$, SZA, SAA | $\Phi_k$ | To test the synergy between reflectances and solar angles to predict $\Phi_k$. |

## 3. Results

### 3.1. Relationships between canopy $SIF_y$ and $F_{yieldLIF}$ and their seasonal variations

The results, in Figure 1a, show that the coefficients of determination of the relationships between $SIF_y$ and $F_{yieldLIF}$ were low and varied highly across the season and that the ratio between diffuse PAR and total PAR cannot entirely explain this inter-daily variability. This indicates that at the diurnal scale $SIF_y$ was weakly correlated to $F_{yieldLIF}$. Note that relations between $SIF_y$ and $F_{yieldLIF}$ analysed at hourly timescale (hourly averages) relatively improved their correlation (Supplementary materials Figure S4). At the seasonal scale (daily averages), in Figure 1b, the results show that the $R^2$ between $\overline{SIF_y}$ and $\overline{F_{yieldLIF}}$ was 0.58, indicating that $\overline{SIF_y}$ and $\overline{F_{yieldLIF}}$ were better correlated at the seasonal timescale. The fraction of diffuse to total PAR cannot explain this correlation. Similarly, the seasonal dynamics in $\overline{SIF_y}$ and $\overline{F_{yieldLIF}}$, in Figure 1c, shows a good correspondence. Although, their agreements tend to diverge at some period of the season. Additionally, note that, overall, the magnitude of both variables has considerably decreased from the starting to the end of the given period.





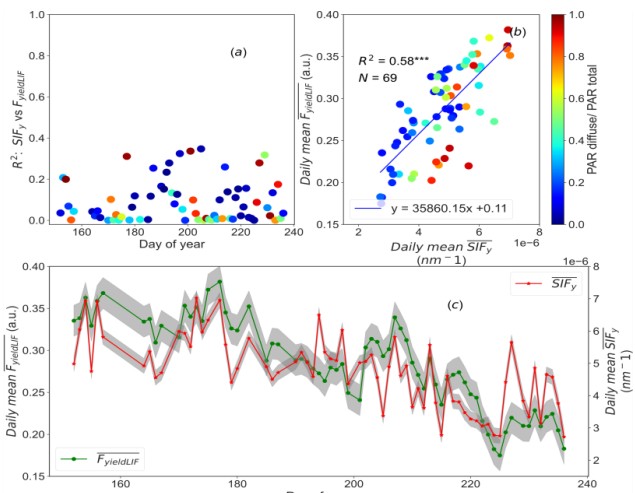

**Figure 1**. Figure 1a shows the inter-daily variations in the coefficient of determination ($R^2$) of the relationship between $SIF_y$ and $F_{yieldLIF}$ and Figure 1b the relationship between daily mean $\overline{SIF_y}$ and $\overline{F_{yieldLIF}}$. In Figures 1a and 1b the colour of the points shows the fraction of diffuse to total PAR with the colour scale on the left of Figure 1b. While Figure 1c depicts seasonal dynamics of $\overline{SIF_y}$ and $\overline{F_{yieldLIF}}$. The shading around the lines indicates the 95% confidence interval. The asterisks stand for the statistical significance level (*** = P ≤ 0.001).

### 3.2. Diurnal variations in PAR, $NIR_v$, Reflectance NIR, $\Phi_k$, SIF, $SIF_y$, and $F_{yieldLIF}$

Figure 2 shows the diurnal cycles (from 9 am to 16 pm) of PAR, $NIR_v$, R-NIR, $\Phi_k$, SIF, $SIF_y$, and $F_{yieldLIF}$.

It shows three sunny and steady weather days and so the PAR constantly increased in the morning to a maximum around noon and decreased in the afternoon for all days. Its values were between 1000 and almost 2000 µmol m$^{-2}$ s$^{-1}$.

The diurnal variations in $NIR_v$ and R-NIR exhibited similar patterns, with the lowest values recorded at noon. The depression observed in $NIR_v$ and R-NIR patterns from 10 am to around 12 pm is attributed to shadows observed within the FOV of the SIF3 instrument as has shown by the sunlit leaves fraction determined from RGB images (Supplementary materials Figure S5 and S6).

$\Phi_k$ surged in the early morning hours (not shown) and then declined from 10 am up to around 12 pm, afterwards, it increased in the afternoon for all days. The depression observed in $\Phi_k$ between 10 am and 12 pm is simultaneous to the decline in $NIR_v$ and in R-NIR. This implies that diurnal dynamics in $\Phi_k$ may be due to the intra-daily pattern in the distribution of sunlit and shaded leaf fraction caused by the geometric relationships between canopy structure and sun's geometry.

It is well-known that diurnal SIF cycles are tightly linked with dynamics in PAR. Conversely, on Figure 2 SIF exhibited different diurnal dynamics for all days than the incident PAR ones. The pattern in SIF declined from 10 am to around 12 pm and was afterwards dominated by dynamics in PAR. It can also be observed that the magnitude of SIF markedly decreased from July, 10th to August, 6th, being from 2.06 to 1.33 mWm$^{-2}$ sr$^{-1}$nm$^{-1}$ (approximately 35% relative decrease in SIF emission).

The diurnal variations in $SIF_y$ surged in the early morning (not shown) and then decreased from 10 am to noon and afterwards increased in the afternoon for the three considered days. Similarly, to SIF, the magnitude of $SIF_y$ also shows an overall decreased from July, 10th to August, 6th. In contrast, the diurnal pattern in $F_{yieldLIF}$ shows a



continuous and significant decrease during the day, with a 10% loss. Note that the range of $F_{yieldLIF}$ have also decreased over the given period. $F_{yieldLIF}$ appears insensitive to the canopy structure and sun-canopy geometry changes, compared to the dynamics in SIF and $SIF_y$, which showed a significant decline in the morning. Besides, it is worth noting that $F_{yieldLIF}$ measurements are continuously recorded (day and night), the full diurnal cycles are presented in Supplementary materials Figure S7.

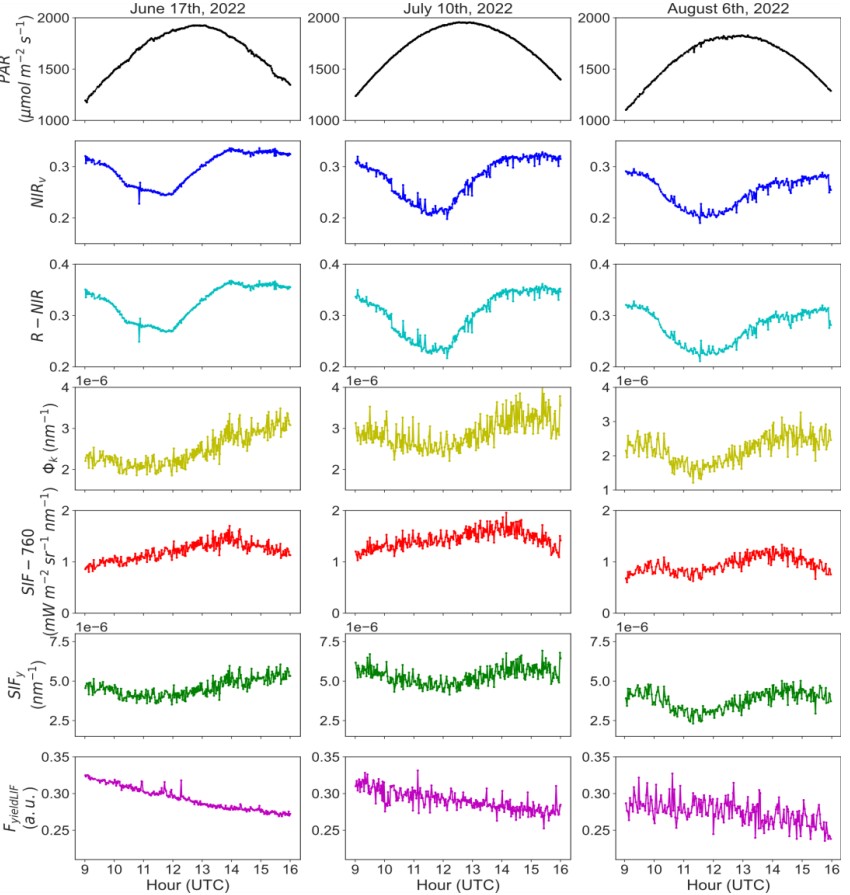

**Figure 2.** Presents the diurnal patterns acquired during three clear sky days of: the diurnal pattern of the photosynthetically active radiation (PAR, *in black* ), the near infrared reflectance of vegetation index (NIR$_v$, *in blue*), the reflectance in the near infrared (R-NIR, *in cyan*), the ratio between SIF$_y$ and $F_{yieldLIF}$ ($\Phi_k$, *in yellow*), the SIF (SIF-760, *in red*), the ratio of SIF over PAR (SIF$_y$, *in green*), and the active chlorophyll fluorescence ($F_{yieldLIF}$, *in magenta*). The data correspond to June 17$^{th}$, July 395  10$^{th}$, and August 6$^{th}$, 2022. The noisy signals observed on July 10$^{th}$ and August 6$^{th}$, 2022 are due to high wind speed with an average value of 2.39 and 3.27 m s$^{-1}$, respectively.

### 3.3. Relationship between $\Phi_k$ and NIR$_v$ and its seasonal variations

Figure 3a shows the R$^2$ of the relationship between NIR$_v$ and $\Phi_k$ at instantaneous scale (acquisition time-step) as a function of the fraction between diffuse and total PAR, while Figure 3b depicts the relationships between $\Phi_k$ and 400  NIR$_v$ at seasonal scale, and Figure 3c underlines their seasonal dynamics.

Conversely, to the weak correlation found between SIF$_y$ and $F_{yieldLIF}$ seen in Figure 1a, the results in Figure 3a show that there are relatively moderate and substantially good relationships between NIR$_v$ and $\Phi_k$ over the season.




Thus, for most of the clear sky condition (ratio diffuse PAR to total PAR < 0.3), $NIR_v$ may explain more than 50% of the instantaneous variations in $\Phi_k$ at the diurnal scale, but the strength of the relationship between these two

variables under clear skies remains variable. The lowest values of $R^2$ are mostly related to diffuse sky conditions. The results in Figure 3b show a weak, but statistically significant relationship between the daily mean $\overline{NIR_v}$ and $\overline{\Phi_k}$ with an $R^2$ of 0.16 at the seasonal scale. This indicates that $\overline{NIR_v}$ is a weak proxy of $\overline{\Phi_k}$ at the seasonal scale. Furthermore, we can also infer that the fraction of diffuse to total PAR explains this correlation, as lower correlation values of $\overline{NIR_v}$ and $\overline{\Phi_k}$ are closely related to clear sky conditions and high correlation values to diffuse sky

conditions. In addition, the seasonal dynamics in $\overline{NIR_v}$ and $\overline{\Phi_k}$ (Figure 3c) exhibited a good match for some days at the seasonal scale. The magnitude of $\overline{NIR_v}$ and $\overline{\Phi_k}$ also varied significantly over the season, which can be caused by rapid changes in ambient environmental conditions and in leaf and canopy biochemical and structural properties. Note that an independent analysis, identical to the one presented here on the relationship between $NIR_v$ and $\Phi_k$, was realised on the relationships between R-NIR and $\Phi_k$. The results shown in Supplementary materials

Figure S8 suggest that the NIR reflectance alone can also be a good proxy of $\Phi_k$ at diurnal timescale. This is paramount for implementing this approach at the satellite scale.

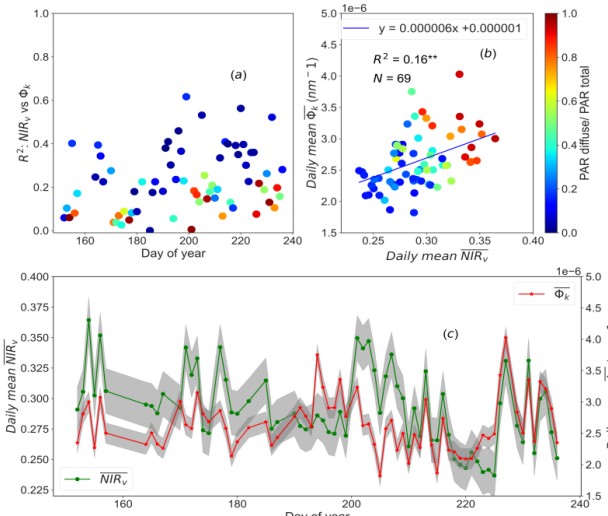

**Figure 3.** Figure 3a exhibits the inter-daily variations of the coefficient of determination ($R^2$) of the relationships between the near-infrared reflectance of vegetation index ($NIR_v$) and the canopy $\Phi_k$ at instantaneous scale, as a function of the fraction

between diffuse and total PAR. Figure 3b presents the seasonal relationship between the daily means $\overline{NIR_v}$ and $\overline{\Phi_k}$, as a function of the fraction between diffuse and total PAR. And Figure 3c shows the seasonal dynamics in $\overline{NIR_v}$ and $\overline{\Phi_k}$. The shaded area indicates the 95% confidence interval. The asterisks stand for the statistical significance level (** = P ≤ 0.01).

### 3.4     Random forest models for predicting $F_{yieldLIF}$ and $\Phi_k$ in a temperate deciduous forest

We tested the potential of RF modelling approach to predict $F_{yieldLIF}$ and $\Phi_k$ based on remotely sensed products.

We intended to show FY-R-SIF$_y$-SA and FY-R-SA models' results for $F_{yieldLIF}$, and $\Phi_k$-R and $\Phi_k$-R-SA for $\Phi_k$ estimates. The other RF models' results for $F_{yieldLIF}$ are given in Supplementary materials Figure S9.

The results show that all random forest models had a strong performance on the prediction of $F_{yieldLIF}$ (Table 2), with OOB $R^2$ varying between 0.86 and 0.90 and adj. $R^2$ between 0.87 and 0.90. In figure 4, the RF models' residuals between observed and predicted $F_{yieldLIF}$ are randomly distributed and $F_{yieldLIF}$ is not over-or under-



estimated. Note that adding SIF (FY-R-SIF, OOB $R^2$ = 0.87 and adj. $R^2$ = 0.88) or $SIF_y$ (FY-R-$SIF_y$, OOB $R^2$ = 0.88 and adj. $R^2$ = 0.89) relatively increases the model performance compared to the FY-R model (FY-R, OOB $R^2$ = 0.86 and adj. $R^2$ = 0.87), but the difference between $R^2$ is not statistically significant. Thus, the use of reflectance bands only allows to predict $F_{yieldLIF}$ and SIF or $SIF_y$ did not provide an additional improvement for predicting $F_{yieldLIF}$ at the acquisition-time step. Substituting SIF for SZA and SAA also showed good model performance (FY-R-SA, OOB $R^2$ = 0.90 and adj. $R^2$ = 0.90). The FY-R-$SIF_y$-SA model revealed a performance similar to the FY-R-SA model's one (FY-R-$SIF_y$-SA, OOB $R^2$ = 0.90 and adj. $R^2$ = 0.90). The predictor importance estimates for FY-R-SA model showed that SZA, SAA, and R410 contribute the most in determining $F_{yieldLIF}$ (Figure 4d), while for FY-R-SIFy-SA model, SZA, R830, SAA, R410, and R740 (Figure 4b) provide the most useful information for $F_{yieldLIF}$ predictions.

**Table 2.** Random forest (RF) model's statistical results for predicting $F_{yieldLIF}$. N denotes the number of data points used for the RF model's testing, adj. $R^2$ represents the adjusted coefficient of determination of the relationship between the test dataset $F_{yieldLIF}$ and the predicted $F_{yieldLIF}$, OOB $R^2$ is the model accuracy on the validation data set (1/3 of the training set), and the RMSE is the root mean square error between observed $F_{yieldLIF}$ and RF model predicted $F_{yieldLIF}$.

| Model | OOB $R^2$ | adj. $R^2$ | RMSE | N |
|---|---|---|---|---|
| FY-R | 0.86 | 0.87 | 0.016 | 1802 |
| FY-R-SIF | 0.87 | 0.88 | 0.016 | 1802 |
| FY-R-$SIF_y$ | 0.88 | 0.89 | 0.015 | 1802 |
| FY-R-SA | 0.90 | 0.90 | 0.014 | 1802 |
| FY-R-SA-$SIF_y$ | 0.90 | 0.90 | 0.014 | 1802 |

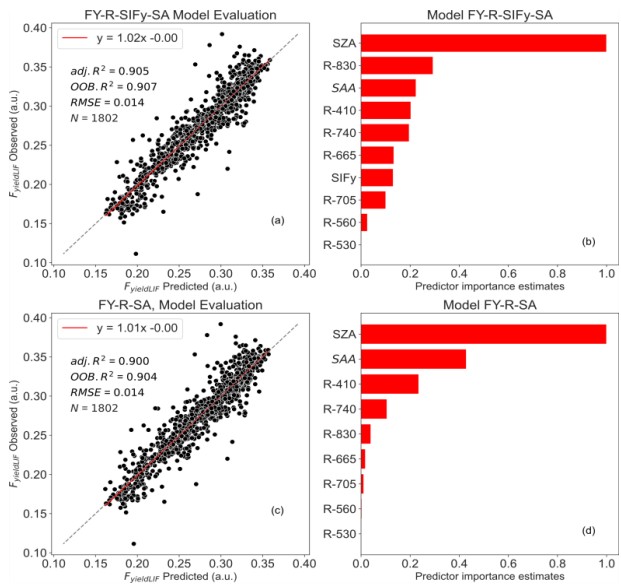

**Figure 4.** Random forest (RF) model outputs: Figure 4a depicts the FY-R-$SIF_y$-SA model performance between observed and predicted $F_{yieldLIF}$, Figure 4b represents the predictor importance estimates for FY-R-$SIF_y$-SA model, Figure 4c represents the FY-R-SA model performance between observed and predicted $F_{yieldLIF}$, and Figure 4d represents the predictor importance estimates for FY-R-SA model. N denotes the number of data points used for the RF model's testing, adj. $R^2$ represents the adjusted coefficient of determination of the relationship between the test dataset $F_{yieldLIF}$ and the predicted $F_{yieldLIF}$, OOB $R^2$ is the model accuracy on the validation data set (1/3 of the training set), and the RMSE is the root mean square error between observed $F_{yieldLIF}$ and RF model predicted $F_{yieldLIF}$. The dashed diagonal line depicts the 1:1 line. FY-R-$SIF_y$-SA denotes $F_{yieldLIF}$ prediction using R, $SIF_y$ and solar angles as inputs; and FY-R-SA includes R, SZA, and SAA to predict $F_{yieldLIF}$.





The results reveal that RF models had good performance in predicting $\Phi_k$ (Figure 5). The best performing model was achieved using R and sun angles as inputs ($\Phi_k$-R-SA, OOB $R^2 = 0.69$ and adj. $R^2 = 0.70$), while R alone explained 58% of $\Phi_k$ on the validation dataset and 62% on the test dataset ($\Phi_k$-R, OOB $R^2 = 0.58$ and adj. $R^2 = 0.62$). The predictor importance estimates (Figures 5b & 5d) show that R410, R740, R665, R705, SZA, and SAA are the main predictors for estimating $\Phi_k$, underlining the dependency of $\Phi_k$ to shadow effects.

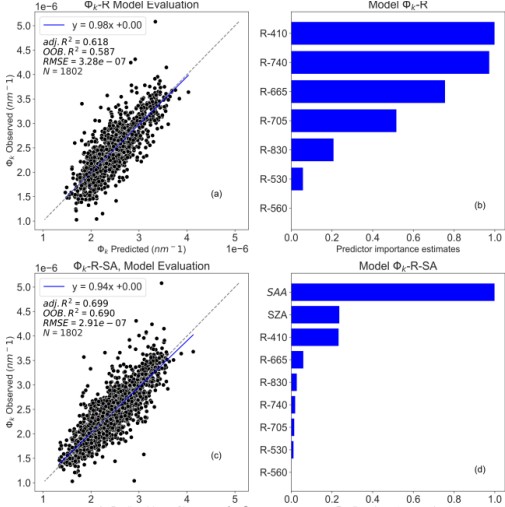

**Figure 5.** Random forest (RF) model outputs: Figure 5a depicts the $\Phi_k$-R model performance between observed and predicted $\Phi_k$, Figure 5b presents the predictor importance estimates for $\Phi_k$-R model, Figure 5c represents the $\Phi_k$-R-SA model performance between observed and predicted $\Phi_k$, and Figure 5d presents the predictor importance estimates for $\Phi_k$-R-SA model. N denotes the number of data points used for the RF model's testing, adj. $R^2$ represents the adjusted coefficient of determination of the relationship between observed and predicted $\Phi_k$, OOB $R^2$ is the model accuracy on the validation dataset (1/3 of the training set), and the RMSE is the root mean square error between observed and RF model predicted $\Phi_k$. The dashed diagonal line depicts the 1:1 line. $\Phi_k$-R denotes $\Phi_k$ prediction using only R; and $\Phi_k$-R-SA integrates R, SZA, and SAA to estimate $\Phi_k$.

## 4. Discussion

### 4.1. Relationships between $SIF_y$ and $F_{yieldLIF}$ at instantaneous and daily timescales

The first objective of this study was to show the effects of canopy structure on SIF signal. The relationship between $SIF_y$ and $F_{yieldLIF}$ was investigated at the daily and seasonal timescales during the growing season from June to August. The results demonstrated that $SIF_y$ and $F_{yieldLIF}$ were more correlated at the seasonal timescale than at the diurnal timescale. Passive SIF is highly dependent on both the structural and physiological properties of the leaf and canopy (Biriukova et al., 2021; Dechant et al., 2022). At the diurnal timescale, far-red SIF is strongly affected by canopy scattering and by the distribution of sunlit and shaded areas at the top and within the canopy (Dechant et al., 2020; Zhang and Zhang, 2023). This study showed that those factors strongly affected $SIF_y$ (SIF normalized by PAR). Further, as $SIF_y$ was estimated using PAR, but not absorbed radiations, $SIF_y$ estimation did not consider the conditions of radiation extinction within the canopy. Therefore, the canopy structural effects can strongly blur the information on the physiological functioning of the vegetation provided by $SIF_y$, and hence lead to low correlations between $SIF_y$ and $F_{yieldLIF}$. Thus, interpreting $SIF_y$ signal for inferring vegetation physiology at the diurnal scale should be carried out with great care, considering the effects of canopy structure and the complex





interactions between structure and illumination geometry. The development of new methods and models are warranted to better explore the possibility to use SIF as a proxy for vegetation functioning at high frequency (seconds to minutes), especially when the vegetation structure is complex and heterogenous, such as in forest stands. On the other hand, the better correlation found at the seasonal timescale can be explained by a potential

removal of short-term changes in illumination conditions, canopy structure, and sun-canopy geometry. Note that the seasonal variability of $\overline{SIF_y}$ is also driven by the seasonal changes in leaf biochemical properties and solar zenith and azimuth angles. These factors can also drive the seasonal dynamics in $\overline{F_{yieldLIF}}$, leading to a better correlation. This may explain why the fraction of the diffuse to total PAR could not entirely explain the relation between $\overline{SIF_y}$ and $\overline{F_{yieldLIF}}$ (Figure 1b). In summary, our results underlined that it is difficult to decouple

vegetation structural and physiological effects in SIF, owing to fluctuations of sun-canopy geometry throughout the day at the diurnal timescale and the difficulties link to the accurate estimation of total SIF and the fraction of absorbed PAR at the canopy scale (Chang et al., 2021).

### 4.2. Effects of canopy structure and sun-canopy geometry on diurnal dynamics in SIF, NIR$_v$, R-NIR, $\Phi_k$, SIF$_y$, and F$_{yieldLIF}$

The fraction of sun absorbed radiation by the canopy (fAPAR) and the fraction of emitted chlorophyll fluorescence that reach the sensor heavily impact SIF. The results obtained during clear sky days revealed that NIR$_v$, R-NIR, and $\Phi_k$ exhibited similar diurnal patterns. This suggests that the diurnal variations in NIR$_v$ that is the product of NDVI and R-NIR, and $\Phi_k$ that represent the product of fAPAR and f$_{esc}$, were dominated by the bidirectional NIR reflectance effect as it has been shown in (Chang et al., 2021). These authors pointed out that the diurnal dynamics

in NIR$_v$ was determined by the diurnal pattern of the reflectance in the NIR within maize crop rows that were under shadow conditions at midday. Sun et al. (Sun et al., 2023a) clearly stated that the dynamics of the fluorescence escape fraction (f$_{esc}$) in homogeneous C3 crop canopy appears to exhibit a diurnal pattern similar to directional reflectance. Further, at intra-daily timescale, $\Phi_k$ is likely to be driven by canopy structure (shadow, leaf angle distribution, etc.) and sun-canopy geometry (SZA and SAA) effects, in particular the distribution of fractions of

sunlit and shaded leaves. This situation can lead to large variability of the diurnal patterns in NIR$_v$ and $\Phi_k$ as has been shown in Figure 2.

The results also highlighted that, at diurnal timescale, the peaks in SIF and PAR do not match (Figure 2), which is probably due to the effects of sun-canopy geometry. Indeed, directionality effects can induce variations in the fraction of sunlit and shaded leaves within the FOV, modulating the actual amount of radiation reaching the leaves

(different from the incident radiation measured at the sensor, unaffected by shading) and therefore affecting canopy total SIF emission. This finding is in contradiction with several studies that showed that the diurnal patterns in SIF is mainly dominated by PAR (Campbell et al., 2019; Wang et al., 2021), but in agreement with (Nichol et al., 2019), who showed that the peaks of PAR and SIF did not match in a Boreal Scots pine canopy. Further, note that at high incident PAR, the light energy might exceed the capacity of photosynthesis. In this case, the plant

photoprotective mechanism known as non-photochemical heat dissipation is activated, leading first to stomatal closure, and hence to SIF emission reduction (Jonard et al., 2020; De Cannière et al., 2022).

The results also showed that the diurnal dynamics in SIF$_y$ and F$_{yieldLIF}$ did not match (Figure 2). This is probably due to both physiological and canopy structure effects. The early decline of SIF$_y$ before noon could be likely due not only to shadow effects, but also to the fact that the PAR was high. These findings corroborate with previous



studies (Loayza et al., 2023; Li et al., 2020; Moya et al., 2019). For instance, Loayza et al. (2023) found that under clear sky conditions, at the diurnal timescale, within potato plants, firstly the chlorophyll fluorescence yield declined drastically when the PAR reached values higher than 1000 µmol m$^{-2}$ s$^{-1}$, and secondly F$_{yieldLIF}$ continuously decreased for PAR > 600 µmol m$^{-2}$ s$^{-1}$. Thus, the continuous decline of F$_{yieldLIF}$ observed here (Figure 2) is likely caused by the fact that the PAR was higher than 1250 µmol photon m$^{-2}$ s$^{-1}$ for the chosen days. Within

this situation, the vegetation photosynthetic capacity could be overwhelmed and the energy-dependent and non-energy-dependent non-photochemical heat dissipation can be triggered. Note that energy-dependent heat dissipation can last from a few seconds to a few minutes, while non-energy-dependent heat dissipation can lead to photoinhibition or photobleaching and can last longer (hours to weeks) (Porcar-Castell et al., 2014). Both mechanisms can induce a decrease in SIF$_y$ and F$_{yieldLIF}$ at the diurnal timescale.

4.3.    **Relationships between NIR$_v$ and Φ$_k$ at daily and seasonal timescales**

Strong correlations were found between NIR$_v$ and Φ$_k$ at the diurnal timescale. However, their correlations varied largely depending on the ratio of diffuse to total PAR, with high correlation corresponding to clear sky conditions and low correlation to diffuse sky conditions. This result suggests that under clear sky conditions NIR$_v$ is relatively a good proxy of Φ$_k$ and hence can be used to take canopy structure and sun-canopy geometry (i.e. crown shadow,

reabsorption, and scattering within leaves and canopies) effects on SIF at the diurnal timescale into account. Indeed, with diffuse sky conditions, canopy structure, shadows and sun-canopy geometry play a minor role in the variations in NIR$_v$ and Φ$_k$, even though there are still strong fluctuations in incident light; justifying the low correlations observed between NIR$_v$ and Φ$_k$ during diffuse sky conditions. On the other hand, the positive weak but statistically significant correlation found between daily mean $\overline{NIR_v}$ and daily mean $\overline{Φ_k}$ at the seasonal timescale

indicates that $\overline{NIR_v}$ and $\overline{Φ_k}$ relations were driven by the fraction between diffuse and total PAR. Indeed, this underlined well NIR$_v$ usage because it was meant to correct reabsorption and scattering effects on SIF at daily and seasonal timescales (Badgley et al., 2017).

4.4.    **Random forest models for F$_{yieldLIF}$ and Φ$_k$ predictions**

How we can determine and properly disentangle the confounding factors, including structural, biophysical, and

physiological canopy components that all contribute to remotely sensed SIF remains a challenging task. SIF has emerged as a promising tool for determining and characterizing structural and physiological vegetation traits. However, the relationships between these confounding factors and SIF are often complex and site-specific and thus require a model with a set of parameters incorporating these complexities. Therefore, in this study, we examined the potential of RF modelling approaches to predict F$_{yieldLIF}$ and Φ$_k$ based on different remotely sensed

input variables under clear sky conditions.

For F$_{yieldLIF}$, the RF models can explain between 86 and 90% of the variability in F$_{yieldLIF}$ (see Table 2 and Figure 4), sustaining that directional reflectance, chlorophyll fluorescence, and sun-canopy geometry (SZA and SAA) can effectively capture relevant variations in F$_{yieldLIF}$. For instance, FY-R-SA and FY-R-SIF$_y$-SA models' predictor importance estimates showed that SZA, SAA, R410, R740, and R830 provide the most useful information for

F$_{yieldLIF}$ predictions (Figure 4). The reflectance in the blue spectral band (R410) is largely affected by the strong blue light absorption by the chlorophyll pigments and it is highly subjective to leaves or canopy shadow conditions, while reflectance in the red-edge (R740) and near-infrared bands characterize the leaf area index and the



chlorophyll content of the entire forest (Zeng et al., 2022b). The red-edge region is mainly used to determine leaf and canopy chlorophyll contents. Because of these abovementioned characteristics of R, it is not surprising that

the combination of reflectance at specific spectral bands can be used to infer effective and relevant information that allow capturing $F_{yieldLIF}$ variations. The data also revealed that adding SIF or $SIF_y$ as predictors did not significantly improve the model performance estimates as it has been shown in (Balde et al., 2023). This result indicates that even at high temporal resolution the contribution of SIF or $SIF_y$ is important compared to each reflectance band individually, but the combined effect of reflectance bands could mitigate or hide the use of SIF

as vegetation physiological proxy. The results showed that SZA and SAA significantly improved the model prediction for $F_{yieldLIF}$ (FY-R-SA). First, the contribution of SZA can be attributed to the illumination conditions because incoming radiations are tightly related to SZA. Second, the effect of SAA is attributable to the anisotropy in reflectance and canopy structure in the azimuthal plane.

For $\Phi_k$, results indicate that RF models can explain between 60 and 70% of the variability in $\Phi_k$ (Figure 5a and

5b). The unexplained 30 or 40% in $\Phi_k$ variance evidenced that the ratio $SIF_y$ over $F_{yieldLIF}$ strongly varies and depends on several factors, including canopy structure, sun geometry, and illumination conditions. Therefore, this suggests that mechanistic models that used $NIR_v$ to approximate the product of fAPAR and $f_{esc}$ are simplistic and do no fully account for the complex interactions between incident radiation and canopy structure, notably due to the distribution of light and shaded leaves at the top and inside of the forest canopy.

**5.        Conclusion**

In this work, the simultaneous and continuous active and passive measurements of chlorophyll fluorescence at the canopy scale in a sessile oak mature forest allowed to analyse the diurnal cycles of key variables, including SIF, $SIF_y$, $NIR_v$, and $F_{yieldLIF}$. A novel remote sensing indicator, $\Phi_k$, the ratio between $SIF_y$ and $F_{yieldLIF}$, which is also theoretically the product of fAPAR and $f_{esc}$, was introduced. On one hand, the relationship between $SIF_y$ and $F_{yieldLIF}$

was evaluated, and on the other hand, the relation between $NIR_v$ and $\Phi_k$ was examined at daily and seasonal scales. Further, several random forest models with reflectances, SIF, and sun angles as inputs were also used to not only predict $F_{yieldLIF}$ and $\Phi_k$, but also to provide sensitivity analysis and interpretation of the model outputs.

The results showed that SIF signal is highly impacted by the canopy structure and the sun-canopy geometry effects, as underlined by the weak correlations found between $SIF_y$ and $F_{yieldLIF}$ at diurnal timescale using instantaneous

measurements. However, $SIF_y$ captured the seasonal dynamics of $F_{yieldLIF}$ by explaining 58% of the variations in $F_{yieldLIF}$. The results also revealed that $NIR_v$ and reflectance at near-infrared (R-NIR) are good proxies of $\Phi_k$ at the diurnal timescale, while their correlations diverged at the seasonal scale.

Based on random forest models, the combination of reflectance, chlorophyll fluorescence, and sun geometry (SZA and SAA) allow to predict $F_{yieldLIF}$ and $\Phi_k$ at the diurnal timescale under clear sky conditions. For instance, the RF

models were able to explain 86-90% of $F_{yieldLIF}$ variability, and 60-70% of $\Phi_k$ variations were explained. Furthermore, the data also revealed that adding SIF or $SIF_y$ as predictors did not improve much the model performance compared to the reflectance-based model. But the predictor importance estimates showed that SIF and $SIF_y$ provide useful and impactful information in determining $F_{yieldLIF}$. This result indicates that even at high temporal resolution the contribution of SIF or $SIF_y$ is important compared to each reflectance band individually,

but the combined effect of reflectance bands could mitigate or hide the use of SIF as vegetation functioning proxy.



Overall, this study provides insights into understanding the complex and difficult relationship that exists between passive SIF and active chlorophyll fluorescence, and into the use of remote sensing data that are readily accessible at satellite scale (spectral reflectance at 10 nm resolution, sun geometry, and chlorophyll fluorescence) to predict $F_{yieldLIF}$ and $\Phi_k$ at canopy scale.


Code and data availability. The computer codes (MATLAB and Python) used in this study are available upon request from the corresponding author.

Supplement. The supplementary materials related to this manuscript is available as a pdf document.


Author contributions. conceptualization, All co-authors; methodology, H.B., G.H., Y.G. and K.S.; instruments design and development YG, AO, GL, GH; software H.B. and G.H.; validation, H.B.; formal analysis, H.B.; investigation, H.B.; resources, H.B., G.H., Y.G. and K.S.; data curation, H.B. and G.H.; writing-original draft preparation, H.B.; writing reviewing and editing, all co-authors; visualization, H.B.; supervision, K.S., Y.G., G.H., and G.L.; project administration, Y.G. and K.S.; funding acquisition, Y.G. and K.S.; All authors have read and agreed to the final version of the manuscript.


Competing interests. The authors declare that they have no conflict of interest.

Funding. This ongoing PhD work is jointly funded by the "Centre National d'Études Spatiales" (CNES) and ACRI-ST (Toulouse, France) (contract CNES-ACRI-ST-Ecole polytechnique-CNRS n°3425). This work was also supported by CNES through the VELIF project focused on the FLEX mission (contracts 4500073234 and 4500073501), The "Program National de Télédétection Spatiales" (PNTS) across the C-FLEX project and EIT Climate-KIC project via the Agriculture Resilience, Inclusive, and Sustainable Enterprise (ARISE) project (EIT 190733).


Acknowledgements We thank Nicolas Delpierre, Daniel Berveiller, Alexandre Morfin, and Clotilde Pérot-Guillaume for their participation on data acquisition and management at Barbeau forest site.

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
