# Peer review of "Data-based investigation of the effects of canopy structure and shadows on chlorophyll fluorescence in a deciduous oak forest"

_EGUsphere, 2023_

## Author Comment (AC1)

Responses to comments: our replies are all in blue color.

Referee #1:

Dear,
We would like to thank you for taking your time to evaluate our work and foremostly for your interesting and useful comments and questions.
We tried to answer your interesting questions and comments (all answers and changes are in blue color).

The presented work shows a very interesting study based on the analysis of measurements obtained in an eddy covariance flux observation site. It is remarkable the use of field data for the study. The use of other reference measurements is lacking, taking advantage of the fact that the study is located in an experimental area, leaf-level measurements could have been used.

We agree that leaf-level measurements are important to better interpret canopy SIF measurements. However, due to lack of time and technical issues, we could not set up leaf-level measurements. These measurements will be provided in the future as we are highly interested in comparing leaf-level and canopy measurements.

Line 354. "In Figure 1c, shows a good correspondence" It will be desirable to provide a quantitative value, perhaps an error estimate, or the difference between the variables compared (with the data from the $NIR_v$ and the R-NIR in the same graph).

This was a formulation mistake and this part was reformulated (Line 351) as "At the seasonal scale (daily averages), in Figure 1b and 1c, the results show that the $R^2$ between $SIF_y$ and $F_{yieldLIF}$ was 0.58, indicating that $SIF_y$ and $F_{yieldLIF}$ were better correlated at the seasonal timescale".

Understanding that the main topic is the structural effects and shadows, please explain why there were not used measurements of the fraction of vegetation shaded along daily and seasonal periods. In line 370 it is commented that the rbg camera was used to determine the sunlit leaves, but there were no further used to normalize or correlate with other variables to reinforce or discard some assumptions and unknowns exposed. For example, on line 368. "The diurnal variations... determined from the RGB". Or line 355. "The magnitude of both variables... of the given period". Did you try to normalize the values by the SZA, or by the sunlit or shaded vegetation fraction?

In this work, we aimed to study canopy structure and sun geometry effects on ground-based measured SIF and to propose a way to correct these structural effects on the SIF signal. Firstly, we explored the RGB images of the FOV of SIF3 captured on sunny days along the season to estimate the sunlit and shaded leaf areas. These data were used to explain visually the effects of shaded leaves on the diurnal SIF measurements. Secondly, as our RGB images were limited in terms of temporal sampling when upscaled at daily or seasonal scale, we could not make use of these data as inputs in the statistical analysis. Further, we introduced $\varphi_k$, which is a new remote sensing indicator that represents the structure and the sun-canopy geometry effects on the SIF signal (structural component of SIF), and we assumed that for a broad and useful use of SIF signal there is a need to find a remote sensing proxy of $\varphi_k$. This is why reflectances and sun-canopy geometry were used in the Random Forest models to predict $\varphi_k$. in other words, our approach was not to normalize SIF using local measurements, only available for our study site, but rather to try to find a remote sensing proxy that could be used even at the satellite scale.

Further, we also aimed to gain a better understanding of the discrepancies between the measured apparent SIF yield ($SIF_y$) and the chlorophyll fluorescence yield measured by LIF ($F_{yieldLIF}$), and whether these discrepancies can be explained by acquisition conditions and canopy characteristics. Our results show that, at this stage, passive SIF measurements cannot be properly standardised. The development of a standardisation method requires further work.

Lines 377-389. The SIF is correlated with the dynamics of the PAR. Obviously, PAR is one of the main factors, but the photosynthetic surface has to absorb the light. This raises the question of why PAR is used to normalize SIF to obtain SIF yield, without applying any correction factor and assuming that the entire area covered by the FOV is fully illuminated vegetation. (SIFy = SIF/PAR) and no (SIFy = SIF/APAR)

This is an interesting question. However, the reason that PAR was used to normalize SIF in our study has been explained in L118-L123: "In addition, the computation of total absorbed photosynthetically active radiation (APAR) requires measurements of the incident, transmitted, and reflected PAR, which cannot be measured with  satellite or    airborne platforms, and are not always available for ground sites (even those belonging to major carbon flux observation networks, such as the Integrated Carbon Observation System, ICOS). This is the reason why for decades the apparent $\Phi F$ was estimated by normalizing the top-of-canopy SIF signal converted in quanta energy by the incident PAR (Daumard et al., 2012; Goulas et al., 2017)". The limitations related to using PAR to normalize SIF were also discussed in L473-L479: "At the diurnal timescale, far-red SIF is strongly affected by canopy scattering and by the distribution of sunlit and shaded areas at the top and within the canopy (Dechant et al., 2020; Zhang and Zhang, 2023). This study showed that those factors strongly affected SIFy (SIF normalized by PAR). Further, as $SIF_y$ was estimated using PAR, but not absorbed radiations, SIFy estimation did not consider the conditions of radiation extinction within the canopy. Therefore, the canopy structural effects can strongly blur the information on the physiological functioning of the vegetation provided by SIFy, and hence lead to low correlations between SIFy and $F_{yieldLIF}$".

Lines 414-416. If $NIR_v$ and r-NIR give almost the same trends in the results, why do you recommend using $NIR_v$?

In this result (Figure 3) we chose using $NIR_v$ and we show the same analysis using R-NIR instead in Figure S8 because $NIR_v$ is a well-established indicator, as shown in the literature. It is important to note that our study was carried out during the vegetation growing season and during this period NDVI was stable. This could explain why $NIR_v$ and R-NIR had the same trends.

Figure 1 and 3. The letters should be in the same place (e.g., top left of the graph boxes).

The letters in Figure 1 and 3 were put at the top left of the graph.

---

## Author Comment (AC2)

Responses to comments: our replies are all in blue color.

Referee #2:

Dear,
We would like to thank you for taking your time to evaluate our work and foremostly for your interesting and useful comments and questions.
We tried to answer your interesting questions and comments (all answers and changes are in blue color).

General comments:

They find the relative difference between Solar-induced and LED-induced fluorescence in a forest site. The interesting point from the presented RF model is the significance of blue and other visible wavelengths to the explanation of Solar/LED yield relationship($\varphi$k). Since equation 6 also consisted of fAPAR×fesc, the blue band, and other factors might be an alternative approach for fesc prediction, too. One of the questions is how to prove the mechanisms of blue band contribution to shadow fraction from observed data (maybe with monitoring camera data). Another point is reproducibility. A justification for diurnal FyieldLIF is lacking in explanation. The reduction in the afternoon fluorescence with LIF might be linked to those of GPP or leaf-level photosynthesis. If the relationship between FyieldLIF and the Light-Use-Efficiency of GPP is weaker than SIFy, the theoretical point will be unsolved.

We found in our study that the blue band contributes to $\varphi_k$ prediction. $\varphi_k$ is theoretically the product of fAPAR and $f_{esc}$. However, in this study with the available measurements we cannot disentangle fAPAR and $f_{esc}$ signals. From our point of view, disentangling these two variables will require modelling approaches.

In this manuscript, we focused on the effects of canopy structure and sun-canopy geometry on passive and active chlorophyll fluorescence signals. We have also investigated whether these effects can be explained by variables accessible by remote sensing. In a work in progress, we are investigating the link between chlorophyll fluorescence and GPP and abiotic variables. An article on these subjects will be submitted shortly (see also our responses to L29 comment below). However, the links between $F_{yieldLIF}$ and LUE are complex at the diurnal scale. These relationships were not explored in this study. There are references that showed strong relations between $F_{yieldLIF}$ measurements and photosynthesis (Flexas et al., 2002; Schreiber et al., 1983).

Specific comments

>Table1

If the SAA is a variable of degree or radian, those can be increasing clockwise to west. In what kind of case does the sun/shade fraction increase/decrease westward? I guess those are not homogenous canopy bidirectional reflectance assumptions. If the illumination angle should be normalized to the principal plane of excitation light, the cosine of (SAA) can be a more realistic factor. Figure 4 indicates the importance of SZA and SAA in the RF model, and those definitions should be clearly and logically defined.

The FOV of our experiment site has a complex canopy structure that can affect the light repartition within and above the canopy. We agree that we could have used the cosine of the

angle instead of the angle itself. However, Random Forest models handle non-linearities related to input variables computation and correlations between variables. We believe that RF results are easier to interpret by using the angle directly as an input, without prior transformation. The relative importance of input variables is independent from input variables units.

Abstract

>L27: geometry effects compared to FyieldLIF.

The geometry effect on LIF is addressed less in the paper. Is there any effect of shade fraction (Figure S5) before the blue LED flash on FOV? Continuously shaded leaves would react differently to other leaves under flash, and those can cause uncertainty on the Fyield.

Our LIF instrument is based on the PAM technique (but without saturation flashes) ( Schreiber, 1986) where chlorophyll fluorescence is induced by non-actinic LED pulses that allow fluorescence-sensing on dark-adapted, shaded and non-shaded leaves without altering $F_{yield}$ (for more details, see Baker et al., 2008; Moya et al., 2019).

>L29:

Could you briefly explain the implication of fluorescence seasonality? Why decreasing? Does it relate to increasing stress factor or light response to quantum yield which is related to the photochemical system openness?

Many factors can explain the seasonal variations in $F_{yieldLIF}$ and its decreasing trend. Among these factors, we have the plant photoprotective mechanism known as non-photochemical heat dissipation, the decline in chlorophyll pigment content of the leaves, and abiotic conditions such as heatwaves and water stress. In our study, there is no clear evidence that supports this. But, it is worth mentioning that during the heatwaves of summer 2022, notably in mid-June, mid-July and in the beginning of August $F_{yieldLIF}$, SIF and GPP have strongly decreased due to an increase in atmospheric water demand (as mentioned above a new manuscript centred on these questions will be submitted soon- Balde et al. in preparation).

Also, a discussion of L486 mentioned $F_{yieldLIF}$ also explained by leaf biochemical and solar angles. Why solar angle is here even though the author assumes LIF output is free from geometric factors?

This might be a misunderstanding: in L486, it is well mentioned that "the seasonal variability of $SIF_y$ is driven by the seasonal changes in leaf biochemical properties and solar zenith and azimuth angles. These factors" (meaning leaf biochemical properties) "can also drive the seasonal dynamics in $F_{yieldLIF}$". This is what we wanted to express. This last sentence will be reformulated as "The leaf biochemical properties can also drive the seasonal dynamics in $F_{yieldLIF}$, leading to a better correlation".

>L30

R-NIR can be rewritten as R850. A hyphen symbol is sometimes confusing.

Thank you for this remark! R850 will be written as R-NIR in the paper.

>L30: the product of NIR by the normalized difference vegetation index

Grammer correction: The Product of A and B.

The correction is considered.

>L190

As far as I know, the optical system called SIF3 with HR1-T sensor is newly developed. Do you have a plan to publish a more detailed explanation of assembly, function, ability to detect signals, and so on? Also, this paper should include the figures of calibration processes, and calibrated spectra (plot of radiance and wavelengths) from upward and downward irradiance at the start and end of the season. Dark current to signal stability is not shown. There is no evaluation of Signal to Noize Ratio. Also, the retrieval uncertainty of SIF should be assessed among different approaches (e.g., iFLD, SFM, BSF, SVD……) compared with the presented 3FLD. It is recommended to enhance the reliability of the findings (especially on a diurnal variation on the O2A band, e.g., van der Tol et al 2023 RSEvol284,113304).

We completely agree that a detailed description of the SIF3 instrument has to be presented. As this paper could not contain all these details, a dedicated paper is currently under preparation by co-authors .

>L275

Please add the figure of upwelling radiance spectra at 757.86, 760.51, and 770.46 nm.

This is a very interesting proposal. The upwelling radiance and downwelling irradiance will be provided in the paper dedicated to SIF3 description.

>L282

Eq (4) can be $=R850 \times NDVI$. Misspelling?

The R850 will be replaced as R-NIR in the paper.

>L300

Why $\varphi K$?

There is no clear reason to choose the SIF/LIF ratio consisting of phi ($\varphi$) and k. If we look at the previous research on this topic, $\varphi$ has been used for quantum yield. It seems confusing.

$\varphi_k$ represents the contribution of the canopy structure and sun geometry effects on the SIF signal. Further, the $\varphi_k$ allows to retrieve the apparent fluorescence yield of SIF. This is why we call the ratio SIF/LIF $\varphi_k$.

>L 555

Any references to blue band contributions?

Zeng et al., 2022b

>Supplementary

Figure S7 shows FyieldLIF is decreasing from morning to afternoon, and the author explained it is derived by activation of dissipation on leaf scale. How could you explain why those are independent of the canopy structural effect? As is shown in Fig S5, the diurnal sun rotation would affect the fraction of sunlit leaves when the instrument was targeting heterogeneous canopy objects. I doubt the diurnal variation of LIF is also a variable of the sunlit fraction, rather than simply explained by hemispherical integrated PAR, especially on a clear sunny day. Thus, additional analysis for the sunlit fraction of LIF would help to minimize uncertainty on target mismatch.

The continuous decline in $F_{yieldLIF}$ from morning to afternoon could be sustained by the activation of the non-photochemical quenching for the dissipation of the excess light energy induced by the high level of incoming radiation. This assumption should be demonstrated by leaf-level measurements that we are highly interested to explore in the future. The shaded leaves fraction could also have an indirect effect on $F_{yieldLIF}$ via photosynthesis, but this effect would be minor compared to the one due to the non-photochemical heat dissipation.